# UAV Image Stitching Based on Optimal Seam and Half-Projective Warp

**Jun Chen** [1,2,3], **Zixian Li** [1,2,3], **Chengli Peng** [4,*], **Yong Wang** [5] **and Wenping Gong** [6]

1    School of Automation, China University of Geosciences, Wuhan 430074, China; junchen@cug.edu.cn (J.C.); lzx2016@cug.edu.cn (Z.L.)
2    Hubei Key Laboratory of Advanced Control and Intelligent Automation for Complex Systems, Wuhan 430074, China
3    Engineering Research Center of Intelligent Technology for Geo-Exploration, Ministry of Education, Wuhan 430074, China
4    Electronic Information School, Wuhan University, Wuhan 430072, China
5    School of Mechanical Engineering and Electronic Information, China University of Geosciences, Wuhan 430074, China; wangy@cug.edu.cn
6    Faculty of Engineering, China University of Geosciences, Wuhan 430074, China; wenpinggong@cug.edu.cn
*    Correspondence: pengcl@whu.edu.cn; Tel.: +86-150-7117-1951

**Abstract:** This paper introduces an Unmanned Aerial Vehicle (UAV) image stitching method, based on the optimal seam algorithm and half-projective warp, that can effectively retain the original information of the image and obtain the ideal stitching effect. The existing seam stitching algorithms can eliminate the ghosting and blurring problems on the stitched images, but the deformation and angle distortion caused by image registration will remain in the stitching results. To overcome this situation, we propose a stitching strategy based on optimal seam and half-projective warp. Firstly, we define a new difference matrix in the overlapping region of the aligned image, which includes the color, structural and line difference information. Then, we constrain the search range of the seam by the minimum energy, and propose a seam search algorithm based on the global minimum energy to obtain the seam. Finally, combined with the seam position and half-projective warp, the shape of the stitched image is rectified to keep more regions in their original shape. The experimental results of several groups of UAV images show that our method has a superior stitching effect.

**Keywords:** UAV image stitching; optimal seam; half-projective warp

## 1. Introduction

With the development of UAV remote sensing technology, its research has been extensively used in urban building planning [1], resources and environment detection [2,3] and other fields. UAV remote sensing has the characteristics of high image resolution, low cost and strong flexibility. It is suitable for collecting low-altitude, high-resolution remote sensing images [4]. In addition to obtaining common RGB images, UAV image remote sensing can also obtain hyperspectral images. Hyperspectral images can provide more spectral information than RGB images [5]. However, due to the limitation of flight altitude, it is difficult for UAV remote sensing to obtain large-area observation images [6]. Therefore, it is necessary to stitch the obtained remote sensing images to improve the information acquisition ability of remote sensing images [7,8].

Image stitching can usually be classified into two main categories. One is the alignment method, which achieves the goal of stitching by accurately aligning the images. The other is seam cutting, which cuts the image by finding a seam with the smallest difference. Advanced image stitching technology can solve the stitching task in most scenes.

The main task of the alignment method is to establish an accurate alignment model [9,10]. Global homography is a common alignment model, and its representative algorithm is Autostitch [11]. This alignment model establishes the corresponding relationship between images through scale-invariant feature transform (SIFT) [12] feature points to

realize image alignment. In order to obtain more accurate alignment ability, some studies use multiple homography [13,14]. As-projective-as-possible warp (APAP) [15] proposes an alignment model based on mesh deformation, which greatly enhances the alignment ability. The alignment ability can also be improved by removing incorrect feature matching [16–21]. In addition, combining line features and point features also has an effect on improving the alignment ability [22,23]. Insufficient alignment ability often leads to ghosting and blurring on the stitched image.

The optimal seam algorithm is an effective solution to eliminate ghosting. The optimal seam refers to finding a seam with the smallest difference in the overlapping area of two images, which should also meet human visual perception and avoid passing through structural objects as much as possible. The whole algorithm is divided into two steps: defining the cost of differences and searching the seams. Defining the precise difference cost can restrict the search range of seams and indirectly improve the effectiveness of seams [24–26]. Most algorithms use the combination of color difference and gradient difference [27].

The stitched image may suffer from lighting inconsistency and other issues. Appropriate fusion technology can reduce exposure differences [28]. A variety of image fusion methods have been used for image stitching, such as mobile phone panorama [29] and UAV image stitching [30]. In addition, many studies focus on obtaining stitched images that are more in line with human visual perception. Some researchers improve the visual effect by reducing the angle distortion [31] and adjusting the rotation angle of the image [32]. In addition, distortion caused by image alignment can also be reduced by adding similarity constraints [33].

In this paper, we propose an image stitching strategy based on the optimal seam algorithm and half-projective warp to solve the image stitching task of UAV with parallax. Firstly, we use global homography to obtain the aligned image. Then, a new difference matrix is defined according to the overlapping part of the aligned image, and a seam search algorithm based on global energy minimization is designed. Finally, according to the position of the seam, we divide the overlapping area of the aligned image, and combine the half-projective warp to obtain a more aesthetic stitching effect.

The contributions of this work involve the following three aspects:

1.  We propose a new optimal seam algorithm and define a new difference matrix, including color, structure and line differences. It can better reflect the difference degree of overlapping regions.
2.  We use the minimum energy to constrain the difference matrix to further limit the search range of the seam, and design a seam search algorithm based on the minimum global energy, which can improve the probability of the seam avoiding structural objects.
3.  According to the position of the seam, we use half-projective warp to correct the image shape, so that more areas maintain the original shape and the stitching effect is improved.

## 2. Related Works

An overview of image stitching and previous posting can be found in [34]. Autostitch [11] is a classical algorithm using the global homography alignment model. It describes the correspondence between two images by detecting feature points between images and calculating a homography. Autostitch must satisfy the requirement that the input images are parallax-free. Otherwise, ghosting and blur will occur due to insufficient alignment ability.

A lot of work has been done to obtain more accurate alignment methods. Dual topography warping (DHW) [13] divides the scene into two planes and aligns them with two homography matrices. Lin et al. [14] used multiple reflection transformations to improve the alignment ability, which can overcome some slight ghosting problems. APAP [15] proposed to divide the image into dense grids, and each grid corresponds to a homogra-

phy matrix, which greatly enhances the image alignment ability. Robust Elastic Warping (REW) [17] was proposed as a feature refinement model based on Bayesian theory to remove mismatched points in image matching and design a robust deformation function to increase the alignment ability. Yuan et al. [22] gave a set of line segment feature detection and matching methods, combined with point features to align the image, and achieved good results. However, due to the fluctuation of the ground and the movement of the camera, UAV images have large parallax. The viewing angle or distance of the same object on the adjacent UAV images will be different, which makes the alignment method unable to effectively solve the parallax stitching problem of UAV images.

Seam-Driven [27] finds the best seam from limited alignment assumptions according to the predefined seam quality measurement. Liao et al. [35] proposed a new iterative seam estimation method to improve the seam vision effect. Fast and robust seam estimation (FARSE) [25] searches for seams by defining the gray weighted distance and differential gradient domain as the difference cost. Li et al. [36] designed a two-image stitching method based on foreground segmentation. A. Eden et al. [37] proposed a two-step optimal seam algorithm, which can stitch the image smoothly even if there is scene motion and alignment error. For most existing seam algorithms, the seam search is usually realized by combining various optimization algorithms [38–40], and is rarely designed according to the defined difference cost.

Shape-Preserving Half-Projective Warps (SPHP) [31] improved the image appearance of non-overlapping regions by transitioning projection transformation to similarity transformation. However, SPHP cannot solve the parallax problem of overlapping regions. Adaptive As-Natural-As-Possible (AANAP) [32] reduces perspective distortion in non-overlapping regions by linearizing the homography and slowly changing it to global similarity, which improves the natural appearance of the stitching results. Chen et al. [33] used line alignment constraints to determine the angle selection of the transformation matrix, and used local and global similarity constraints to preserve the original shape of the image. However, these algorithms for improving the visual performance are usually based on alignment methods and are not combined with optimal seaming algorithms.

## 3. Materials and Methods

In this section, we detail our proposed algorithm, including the optimal seam algorithm and half-projective warps. Figure 1 demonstrates the overall flow of the algorithm. Firstly, we use global homography to register image 1 and image 2; then, we find an optimal seam and cut two registered images, respectively; we use the original image 1 and the cut image 2 to register using the half-projective warp; finally, we stitch the two images obtained by half-projective warp according to the optimal seam.

### 3.1. Optimal Seam Algorithm

In the registration phase, we first use global homography to construct a warp from the reference image to the target image. Given the input images $I$ and $I'$ along with the corresponding speeded up robust features (SURF) [41] feature matching points $x = [x, y]^T$ and $x' = [x', y']^T$, the linear transformation of homogeneous coordinates between two images can be represented as

$$\tilde{\mathbf{x}}' = \mathbf{H}\tilde{\mathbf{x}}, \tag{1}$$

where $\tilde{\mathbf{x}}$ is $x$ in homogeneous coordinates. $\mathbf{H} \in \mathbb{R}^{3\times3}$ defines the homography. The rows of $\mathbf{H}$ are given by $\mathbf{h}_1 = [h_1, h_2, h_3]$, $\mathbf{h}_2 = [h_4, h_5, h_6]$, $\mathbf{h}_3 = [h_7, h_8, 1]$. The mapping between two images can be written as

$$x' = \frac{h_1 x + h_2 y + h_3}{h_7 x + h_8 y + 1}, \tag{2}$$

$$y' = \frac{h_4 x + h_5 y + h_6}{h_7 x + h_8 y + 1}. \tag{3}$$

Then, we obtain the resultant images by warping the input images with **H** and place them on the same reference plane.

The basis of the optimal seam algorithm is to estimate a seam with the smallest difference from the overlapping regions of two registered images. Then, the two registered images are segmented and reorganized according to the seam. Our proposed method is divided into two steps:

(1) Construct the difference matrix of the overlapping regions of two registered images;
(2) Search for the optimal seam on the difference matrix.

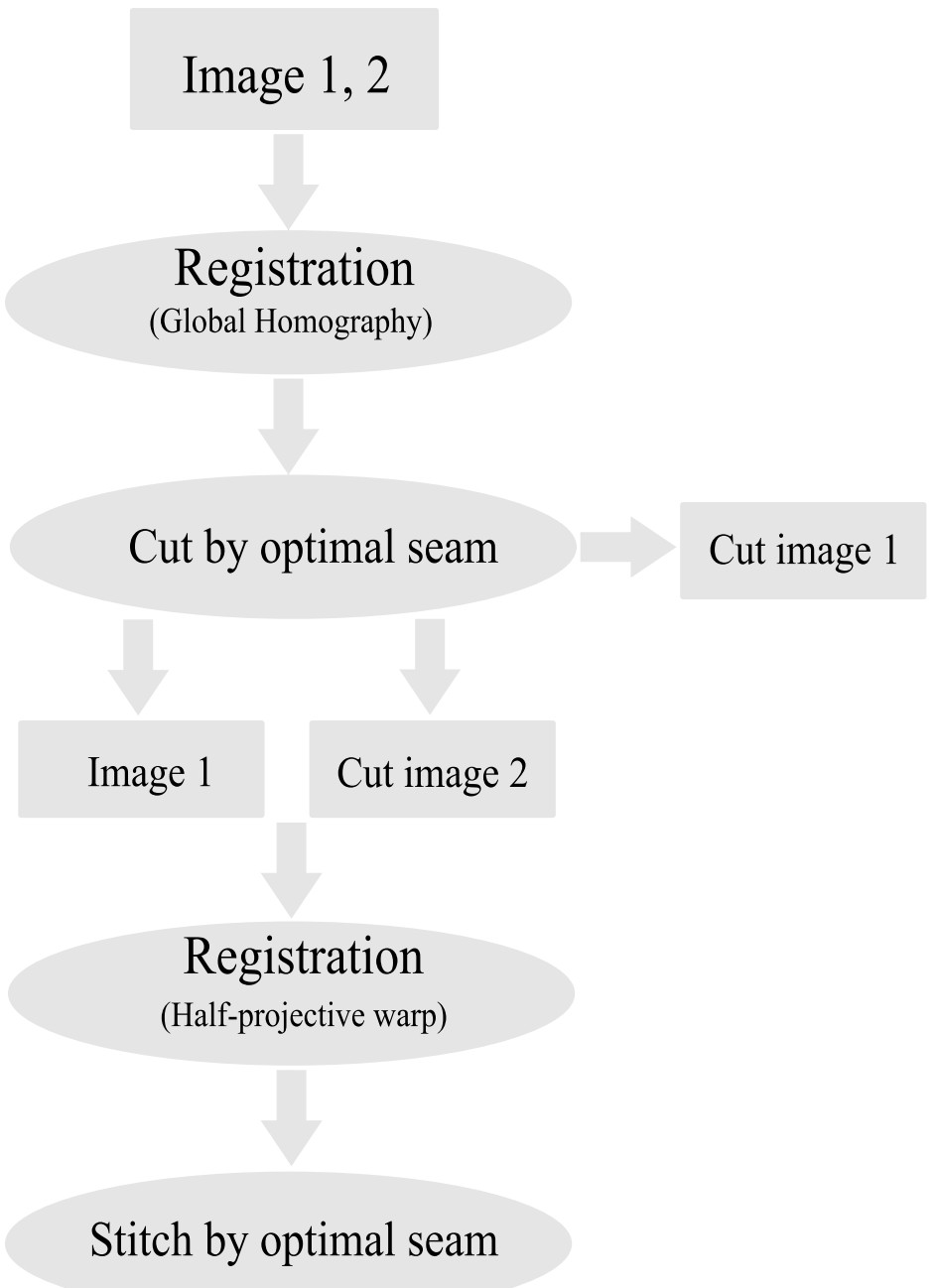

**Figure 1.** Simplified representation of our proposed method.

Firstly, we extract the overlapping regions of two registered images, which are denoted as $\Omega$ and $\Omega'$. Then, we define a difference matrix reflecting the similarity between overlapping regions $\Omega$ and $\Omega'$. Previous algorithms usually use the color difference and structure

difference as the criteria to judge the similarity, and experiments show that this combination of differences is effective. In order to better cater to the human eye's perception of color, we use the color difference in LAB color space to calculate the color difference.

$$\bar{r} = \frac{\Omega_R + \Omega'_R}{2},\tag{4}$$

$$\Delta L = \left(2 + \frac{\bar{r}}{256}\right) \times (\Omega_R - \Omega'_R)^2,\tag{5}$$

$$\Delta A = 4 \times (\Omega_G - \Omega'_G)^2,\tag{6}$$

$$\Delta B = \left(2 + \frac{255 - \bar{r}}{256}\right) \times (\Omega_B - \Omega'_B)^2,\tag{7}$$

$$E_{color} = \sqrt{\Delta L + \Delta A + \Delta B},\tag{8}$$

where $E_{color}$ is the color difference. $\Omega_R$, $\Omega_G$, $\Omega_B$ are the RGB (red, green, blue) channel values of $\Omega$. $\Omega'_R$, $\Omega'_G$, $\Omega'_B$ are the RGB channel values of $\Omega'$.

However, combined with the experiments of UAV images, the errors in the results of the seam algorithm often focus on some structural objects. Due to the large parallax of UAV images, it is difficult to achieve accurate registration of structural objects. If the seam passes through these structural objects, there is a high possibility of visual deviation on the structural objects. We find that some ideal seam paths often follow some roads or grasslands, which generally belong to the low-frequency part of the image. Therefore, we use the high-frequency parts of the overlapping regions to construct the structural difference in order to reduce the structural difference in the low-frequency part. We use Gaussian filtering with parameter $\sigma_1$ for $\Omega$ and $\Omega'$ to obtain $\Omega_1$ and $\Omega_2$. Then, we use Gaussian differential edge detection to calculate the structural difference between $\Omega_1$ and $\Omega_2$.

$$E_{\Omega_i} = \frac{1}{\sqrt{2\pi}}\left(\frac{1}{\sigma_2}e^{-\left(x_i^2 + y_i^2\right)/2\sigma_2^2} - \frac{1}{\sigma_3}e^{-\left(x_i^2 + y_i^2\right)/2\sigma_3^2}\right),\tag{9}$$

$$E_{structure} = E_{\Omega_1} - E_{\Omega_2},\tag{10}$$

where $i = 1, 2$. $E_{structure}$ is the structural difference, $\sigma_2$ and $\sigma_3$ are the difference parameters, and specific values will be mentioned in the experimental section.

In addition, in order to make the difference in structural objects more obvious, we also introduce linear difference. By detecting the line segment information of the object, the seam can avoid passing through the object with straight line edges. In particular, we use the line segment detector (LSD) [42] to obtain the linear information of the overlapping regions $\Omega$ and $\Omega'$ and subtract them to obtain the linear difference, denoted as $E_{line}$.

We add up the above three differences:

$$E = E_{color} + E_{structure} + E_{line},\tag{11}$$

where $E$ is the difference matrix, which is a two-dimensional numerical matrix, as shown in Figure 2. The value of $E$ represents the difference. The start point and end point of the seam are usually at the junction of the two registered images. If two pixels can be connected into an uninterrupted line on the two-dimensional matrix, they must be in the same eight-connected region.

We set a threshold $e$ and limit the difference value of pixels on the seam to be less than $e$. Under the condition that the start point and the end point are in the same eight-connected region, $e$ should be minimized as much as possible. We can sort all the difference values on $E$ and quickly calculate the minimum threshold $e$ by using the binary search algorithm. Under the condition of minimum threshold $e$, the eight-connected region where the start point and the end point are located is denoted as $R$, and the search region of the seam is limited to $R$, as shown in Figure 2. The calculation flow of $R$ is shown in Algorithm 1.

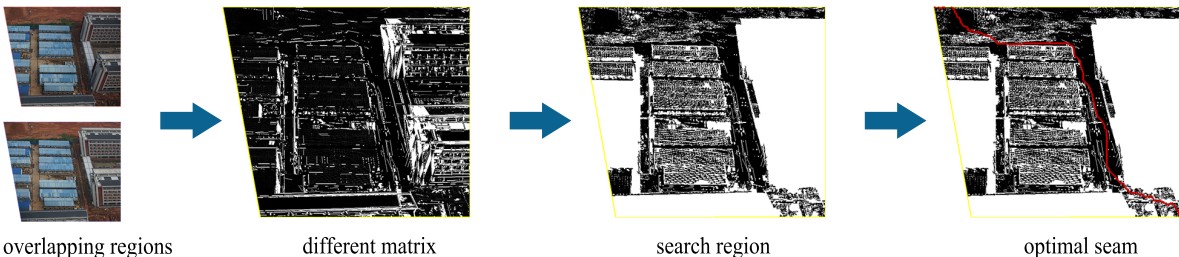

overlapping regions　　　　different matrix　　　　search region　　　　optimal seam

**Figure 2.** Specific simplified process of seam searching algorithm.

---

**Algorithm 1:** Calculation of search region

　　**Input:** Different matrix $E \in \mathbb{R}$, start point $p_1(x_1, y_1)$, end point $p_2(x_2, y_2)$
　　**Output:** search region $R \in \mathbb{R}$
1　Define $e$ as a collection of all elements in $E$;
2　Sort $e$ in ascending order;
3　Define $N$ as the total number of elements of $e$;
4　define $left = 1, right = N, middle = \lfloor \frac{N-1}{2} \rfloor$;
5　**while** $e_{left} < e_{right}$ **do**
6　　　$middle = \lfloor \frac{right-left}{2} \rfloor + left$;
7　　　$R_{middle} = E(E < e_{middle})$;
8　　　**if** $p_1, p_2 \in$ *the same eight-connected region in* $R_{middle}$ **then**
9　　　　　right=middle;
10　　　**else**
11　　　　　left=middle;
12　　　**end**
13　**end**
14　$R = E(E < e_{middle})$;

---

We obtain the search region $R$ through numerical constraints. Each pixel in the search region $R$ has a specific difference value. Next, we propose a seam search algorithm based on the minimum global difference. We begin from the start point and expand in eight adjacent directions, update the pixel difference value in $R$ to the sum of the minimum difference from the starting point, and take the updated pixels as new expansion points until we expand to the end point. Then, we begin from the end point, along the pixel path with the smallest difference sum value, and return to the start point and obtain our optimal seam. The specific simplified process is shown in Figure 3. The position of the optimal seam in the difference matrix is shown in Figure 2, represented by a red line.

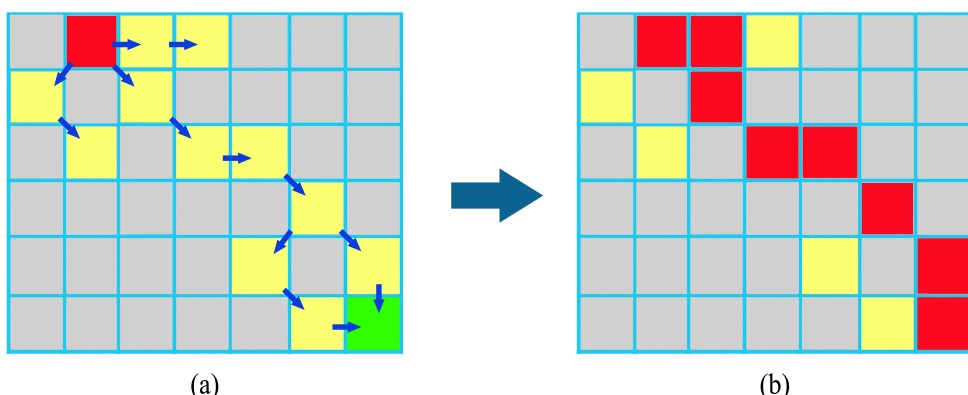

(a)　　　　　　　　　　　　　　　　　　(b)

**Figure 3.** Specific simplified process of proposed seam searching. (**a**) is the brief process of expansion. The start point is red, the end point is green, and the yellow region is the search region $R$. (**b**) The red region is the final seam path.

*3.2. Half-Projective Warps*

The components of the stitched image obtained by the seam algorithm come from the registered images. However, in the registration process of UAV images with large parallax, the non-overlapping region will deform, and the object will have unfriendly shape distortion. These problems remain in the results obtained by the seam algorithm.

In order to solve this problem, we introduce half-projective warps. Homography matrix **H** usually corresponds to the projection transformation, resulting in tensile deformation of the object. Meanwhile, similarity transformation only changes the size and direction of the object, and maintains the original shape. Half-projective warp performs projection transformation in overlapping regions and performs similarity transformation in non-overlapping regions, and there is a smooth transition region between the two transformations.

First, we use $\theta$ to rotate the coordinate system $(x, y)$ to $(u, v)$.

$$\theta = \operatorname{atan2}(-h_8, -h_7), \tag{12}$$

$$x = u\cos\theta - v\sin\theta, \tag{13}$$

$$y = u\sin\theta + v\cos\theta. \tag{14}$$

Substituting Equations (13) and (14) into Equations (2) and (3), the new mapping can be written as

$$H(u, v) = \begin{bmatrix} x' \\ y' \end{bmatrix} = \begin{bmatrix} \frac{\hat{h}_2}{1-cu}v + \frac{\hat{h}_1 u + \hat{h}_3}{1-cu} \\ \frac{\hat{h}_5}{1-cu}v + \frac{\hat{h}_4 u + \hat{h}_6}{1-cu} \end{bmatrix}, \tag{15}$$

where $c = \sqrt{h_7^2 + h_8^2}$, $\hat{h}_1, \hat{h}_2, \hat{h}_3, \hat{h}_4, \hat{h}_5, \hat{h}_6$ are the new constant coefficients. Then, we divide $\mathbb{R}^2$ by the line $u = u_1$ and $u = u_2$ into three spaces, where each space corresponds to a warp. For the whole space, the warping function is defined as

$$w(u, v) = \begin{cases} H(u, v), & u \le u_1 \\ T(u, v), & u_1 < u < u_2, \\ S(u, v), & u_2 \le u \end{cases} \tag{16}$$

$$T(u, v) = \begin{bmatrix} f_1(u)v + f_2(u) \\ f_3(u)v + f_4(u) \end{bmatrix}, \tag{17}$$

$$S(u, v) = \begin{bmatrix} -s_2 v + s_1 u + s_3 \\ s_1 v + s_2 u + s_4 \end{bmatrix}, \tag{18}$$

where $T(u, v)$ is a transition transformation, and $f_1, f_2, f_3, f_4$ are quadratic functions of $u$. $S(u, v)$ is a similarity transformation, and $s_1, s_2, s_3, s_4$ are constant parameters. When $u$ is a constant, $H(u, v)$, $T(u, v)$ and $S(u, v)$ are linear functions about $v$. $H(u, v)$ and $T(u, v)$, $T(u, v)$ and $S(u, v)$ can be continuous in $u = u_1$, $u = u_2$, respectively. Therefore, we have sufficient linear constraints to find $f_1, f_2, f_3, f_4, s_1, s_2, s_3, s_4$.

Next, we give the specific calculation process. When $u = u_1$ and $u = u_2$, according to the continuity of $w(u, v)$, we can obtain the following equations.

$$f_1(u_1) = \frac{\hat{h}_2}{1 - cu_1}, \tag{19}$$

$$f_1'(u_1) = \frac{c\hat{h}_2}{(1 - cu_1)^2}, \tag{20}$$

$$f_1(u_2) = -s_2, \tag{21}$$

$$f_1'(u_2) = 0. \tag{22}$$

Since we have four linear constraints, we can solve four parameters. Where $s_2$ occupies one parameter, $f_1$ is the quadratic function of $u$, with three parameters. By solving the above linear equations, we can obtain $f_1$ and $s_2$. Similarly, we can solve $f_3$ and $s_1$. According to the same strategy, we can also obtain the following equations.

$$f_2(u_1) = \frac{\hat{h}_1 u_1 + \hat{h}_3}{1 - cu_1},\qquad(23)$$

$$f_2'(u_1) = \frac{\hat{h}_1 + c\hat{h}_3}{(1 - cu_1)^2},\qquad(24)$$

$$f_2(u_2) = s_1 u_2 + s_3,\qquad(25)$$

$$f_2'(u_2) = s_1.\qquad(26)$$

Since we have solved $s_1$ and $s_2$, we still have enough linear constraints to solve $f_2$ and $s_3$. Similarly, we can solve $f_4$ and $s_4$.

Next, we describe how to determine the values of $u_1$ and $u_2$ in combination with the position of the seam. We denote the regions retained after cutting as $R_1$ and $R_2$, and the regions removed as $R_1'$ and $R_2'$. We use $R_1$ and the original target image for half-projective warp, and the overlapping regions of the two images are in $R_2'$. For $R_2'$, as the position of the seam will also change according to our warp, $R_2'$ will eventually be removed. We only need to make $R_1$ and $R_2$ undergo a similar transformation as much as possible as a constraint for judging $u_1$ and $u_2$. We use the deviation of warp function $w(u, v)$ from the nearest similarity transformation in the Frobenius norm as a cost.

$$C = \sum_{i=1}^{2} \min_{\alpha_i, \beta_i} \iint_{(x,y)\in R_i} \left\| \begin{bmatrix} \frac{\partial x}{\partial u} - \alpha_i & \frac{\partial x}{\partial v} + \beta_i \\ \frac{\partial y}{\partial u} - \beta_i & \frac{\partial y}{\partial v} - \alpha_i \end{bmatrix} \right\|_F^2 dxdy,\qquad(27)$$

where $C$ is a nonlinear function of $u_1$ and $u_2$, and the positions of $u_1$ and $u_2$ are determined by regularly sampling the parameter space $(u_1, u_2)$.

## 4. Results

In this section, we introduce the experiment of our method on UAV images, and compare it with the existing alignment algorithm and optimal seam algorithm. The test images in the experiment were taken outdoors by the feimaD200 UAV equipped with SONY ILCE-600, including some villages, villas and construction sites; this basically reflects the characteristics of UAV images. In order to verify the effectiveness of our method, our experiments were mainly set up in the following aspects: (1) our seam algorithm and the most advanced alignment algorithm comparison; (2) our seam algorithm and other seam algorithms comparison; (3) combined with the half-projective warp of seam, analyzing the effect of image correction.

In particular, in the filtering step of defining the structural difference, we assign 0.4, 0.6 and 0.8 to $\sigma_1$, $\sigma_2$ and $\sigma_3$ for Gaussian weight, respectively. All the experiments were implemented on a computer with a 2.90 GHz Intel Core i5-10400F CPU and 16-GB RAM.

### 4.1. Visual Comparison

In this section, we compare several popular alignment stitching algorithms to verify the applicability of our stitching algorithm in UAV image stitching. Comparison methods include Autostitch [11], APAP [15], AANAP [32] and REW [17], and the codes were provided by the authors. These algorithms focus on stitching images through accurate alignment. We applied these algorithms to UAV images and compared them with our methods. We tested three stitching cases in the villa area. There is a large translational movement between the test images in Figure 4, resulting in a large parallax. Autostitch, APAP, AANAP and REW present ghosting and blurring, including houses and cars. Some representative areas are indicated with red boxes. REW and APAP are affected by the most

serious artifacts; Autostitch and AANAP are also blurred to varying degrees in complex house structures. On the contrary, our method finds a seam from the overlapping area of the image, which can avoid houses and cars and follow the flat road. Then, the image is segmented and reorganized along the seam. Each object comes from only one image, avoiding ghosting and blurring. Figure 5 shows a case of test images with rotation and translation. It can be seen from the area indicated by the red box that Autostitch, APAP, AANAP and REW all have serious artifacts on the indicated houses. Because the car has different perspectives on the two input images, the car also has ghosting to varying degrees. In Figure 6, AutoStitch and APAP still have some ghosting on the house. AANAP mitigates ghosting on some houses, but reduces the alignment accuracy on some cars. Although REW eliminates ghosting to a certain extent, there is one house that is misplaced. Large parallax makes these algorithms unable to accurately align objects and they produce different degrees of fuzzy ghosting. The experiments show that our method can solve these problems and is more suitable for UAV image stitching.

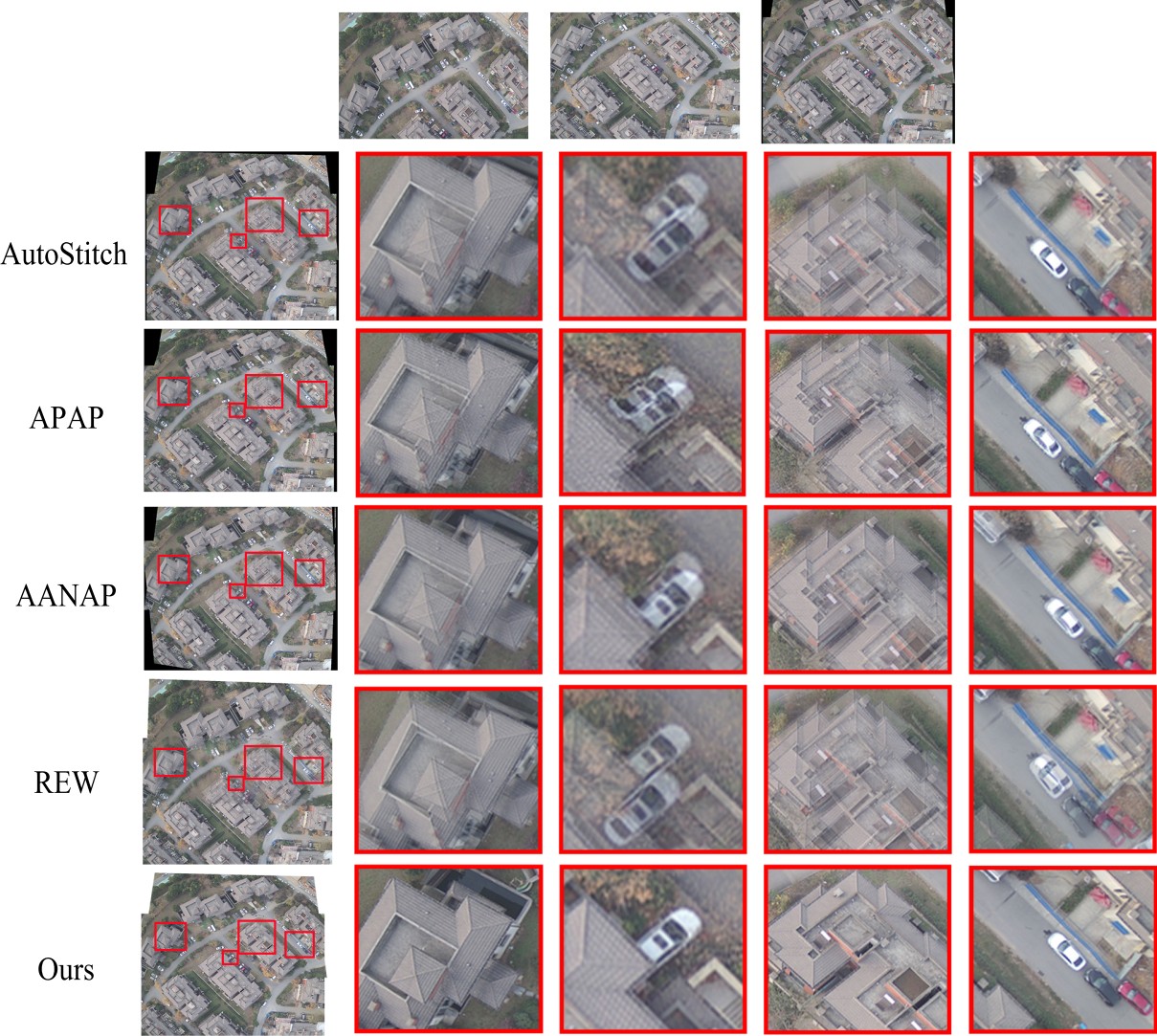

**Figure 4.** Stitching results among various popular methods. The first line is the input images and the position of our seam. The second, third, fourth, fifth and sixth lines are the results of AutoStitch, APAP, AANAP, REW and our seam algorithm. The red boxes highlight some details. The percentage of overlap between two images in this case is 62.98%.

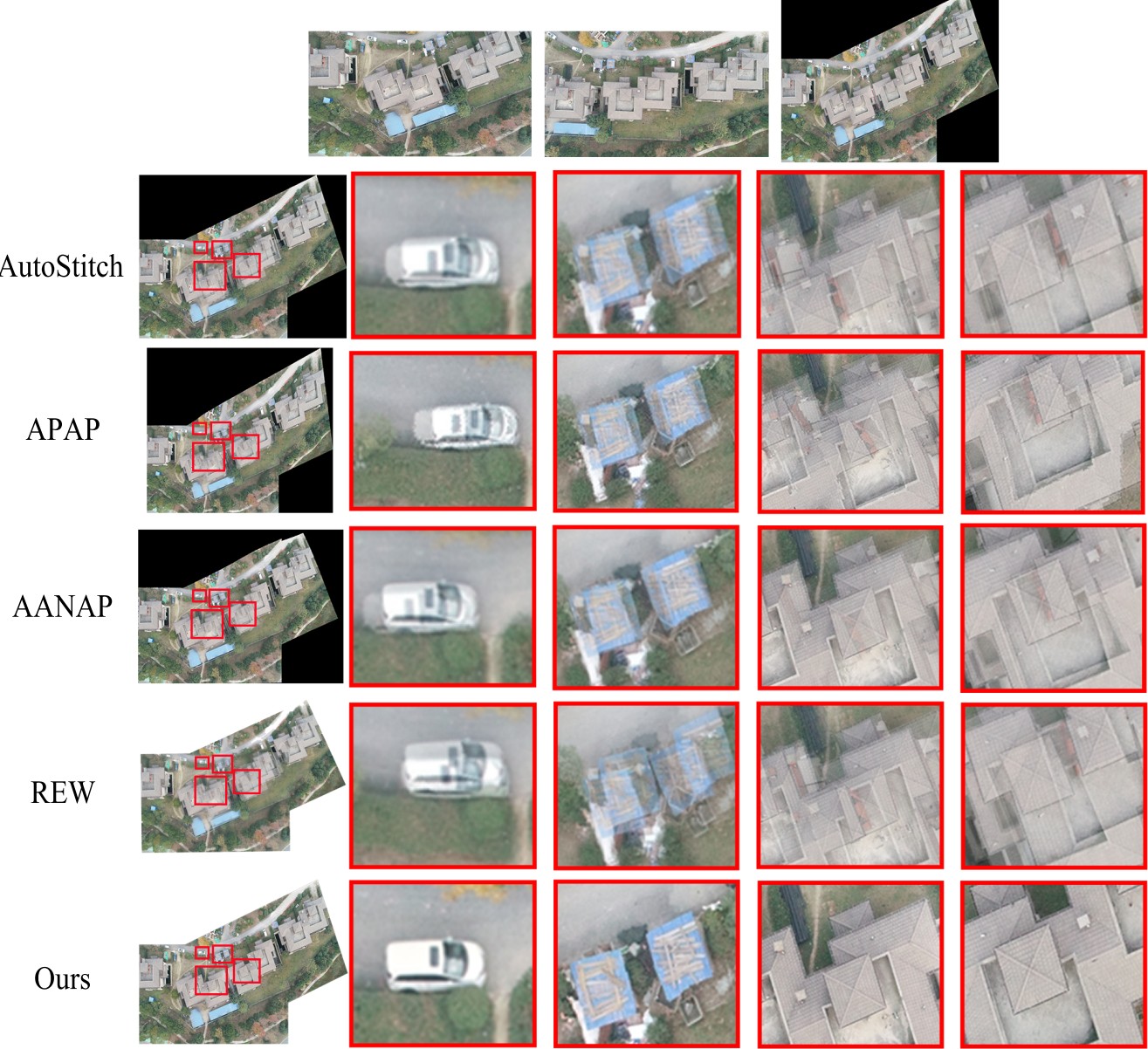

**Figure 5.** Stitching results among various popular methods. The first line is the input images and the position of our seam. The second, third, fourth, fifth and sixth lines are the results of AutoStitch, APAP, AANAP, REW and our seam algorithm. The red boxes highlight some details. The percentage of overlap between two images in this case is 52.74%.

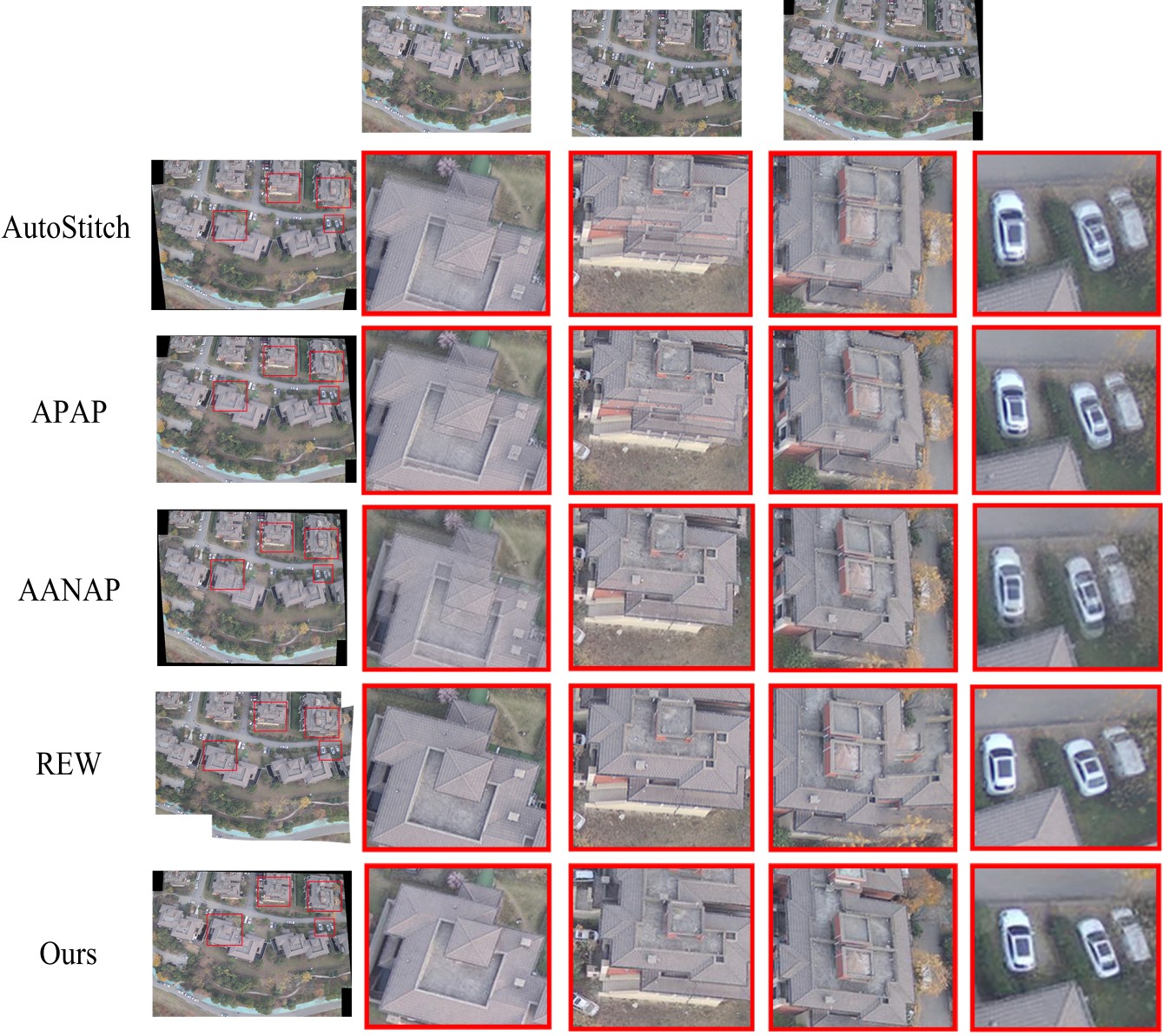

**Figure 6.** Stitching results among various popular methods. The first line is the input images and the position of our seam. The second, third, fourth, fifth and sixth lines are the results of AutoStitch, APAP, AANAP, REW and our seam algorithm. The red boxes highlight some details. The percentage of overlap between two images in this case is 81.40%.

### 4.2. Seam Comparison

In this part, we show the results of different seam algorithms to prove the effectiveness of our method. Specifically, we compare a fast and robust seam estimation (FARSE) [25], perceptual-based seam cutting (PSC) [43] and quality evaluation-based iterative seam estimation (QEISE) [35]. Figure 7 shows the results of four different test images. We indicate the detected seams in red and indicate some houses and buildings with blue boxes. These experiments show that the seam algorithms can eliminate ghosting and blurring. The approximate path of the seam obtained by FARSE always passes through some structural objects. The seams obtained by PSC will appear at the boundary of the overlapping area, resulting in a large area of staggering. QEISE has similar problems, and the path of seams is too tortuous. In contrast, the seams detected by our algorithm are shorter, smoother and pass through flat areas. Seams effectively avoid some unnecessary structural objects, and

meet human visual perception to a greater extent. The experiment proves that our seam algorithm is better than others and can achieve a better stitching effect.

In addition, we quantitatively evaluate the quality of the seams. We use two different objective evaluation criteria to measure the quality of the seams. Firstly, we utilize the peak signal to noise ratio (PSNR) score to measure the similarity degree, and we judge the similarity degree of the seam on the reference image and the target image according to the score. The PSNR score is directly proportional to the similarity degree.

$$Q_{PSNR} = 10 \times \log_{10}\left(\frac{(2^n - 1)^2}{MSE}\right), \tag{28}$$

where $Q_{PSNR}$ is the PSNR score of the seam. $MSE$ is the mean square error of the seam in the corresponding pixels of the reference image and the target image. $n$ is the number of bits of each sampling value, taken as 8.

Then, we calculate the structural similarity (SSIM) score of each pixel of the seam on the reference image and the target image, and then define the quality of the seam as:

$$Q_i = \frac{1 - \text{SSIM}(p_i)}{2}, \tag{29}$$

$$Q_{SSIM} = \frac{1}{N}\sum_{i}^{N} Q_i, \tag{30}$$

where $Q_{SSIM}$ is the quality score of the seam. $p_i$ represents the $i$th pixel on the seam, and $Q_i$ represents the quality score of the ith pixel. $N$ is the sum of pixels of the seam. The SSIM score ranges from $-1$ to 1. The final quality score is inversely proportional to the seam quality.

We use the four stitching cases corresponding to Figure 7. We evaluate the seams obtained by four different methods according to the above measurement criteria. $Q_{PSNR}$ is shown in Table 1 and $Q_{SSIM}$ is shown in Table 2. The seams obtained by PSC often appear at the boundary of the image, and the seams are quite different in the pixels corresponding to the reference image and the target image. Therefore, the $Q_{PSNR}$ of PSC is lower and the $Q_{SSIM}$ is higher. Visual differences often appear on structural objects. Compared with other methods, our seams can better avoid passing through structural objects, so we have higher $Q_{PSNR}$ and lower $Q_{SSIM}$.

In particular, we extract 500 pixels equally spaced from the seam lines of the four seaming algorithms on the four stitching examples, and calculate their quality curves separately for comparison. If there is a large peak in the quality curve at the seam, it indicates that the seam passes through an area of large difference. As shown in Figure 8, since the seams of PSC often appear at the boundary with a large difference, the quality curve of the seam obtained by PSC will have a continuous peak, especially in Case 2 and Case 3. Our seam quality curves have fewer and lower peaks than the others, indicating that our seams pass through smaller differences in paths.

In addition, we compare the time consumed by the stitching algorithms. We compare the time required to find the seam from the registered images. As shown in Table 3, FARSE is aimed at the rapid search of seams, and thus takes less time than other methods. In addition to FARSE, our method consumes less time than PSC and QEISE under the condition of ensuring seam quality.

In general, the seams found by our seam algorithm are usually on flat areas with small differences. However, not all UAV test images have obvious flat areas. Figure 9 shows two special stitching cases. The houses on the two stitching cases are relatively dense, and there is no obvious flat area to connect the start point and end point of the seam. In addition, one end of the seam is on a house so that the seam has to pass through the house. As shown in Figure 9, different seam algorithms cannot avoid passing through the houses indicated with red boxes. However, in the case of dense houses, our seam algorithm can minimize

passing through avoidable houses as much as possible compared with FARSE, as shown by the green box.

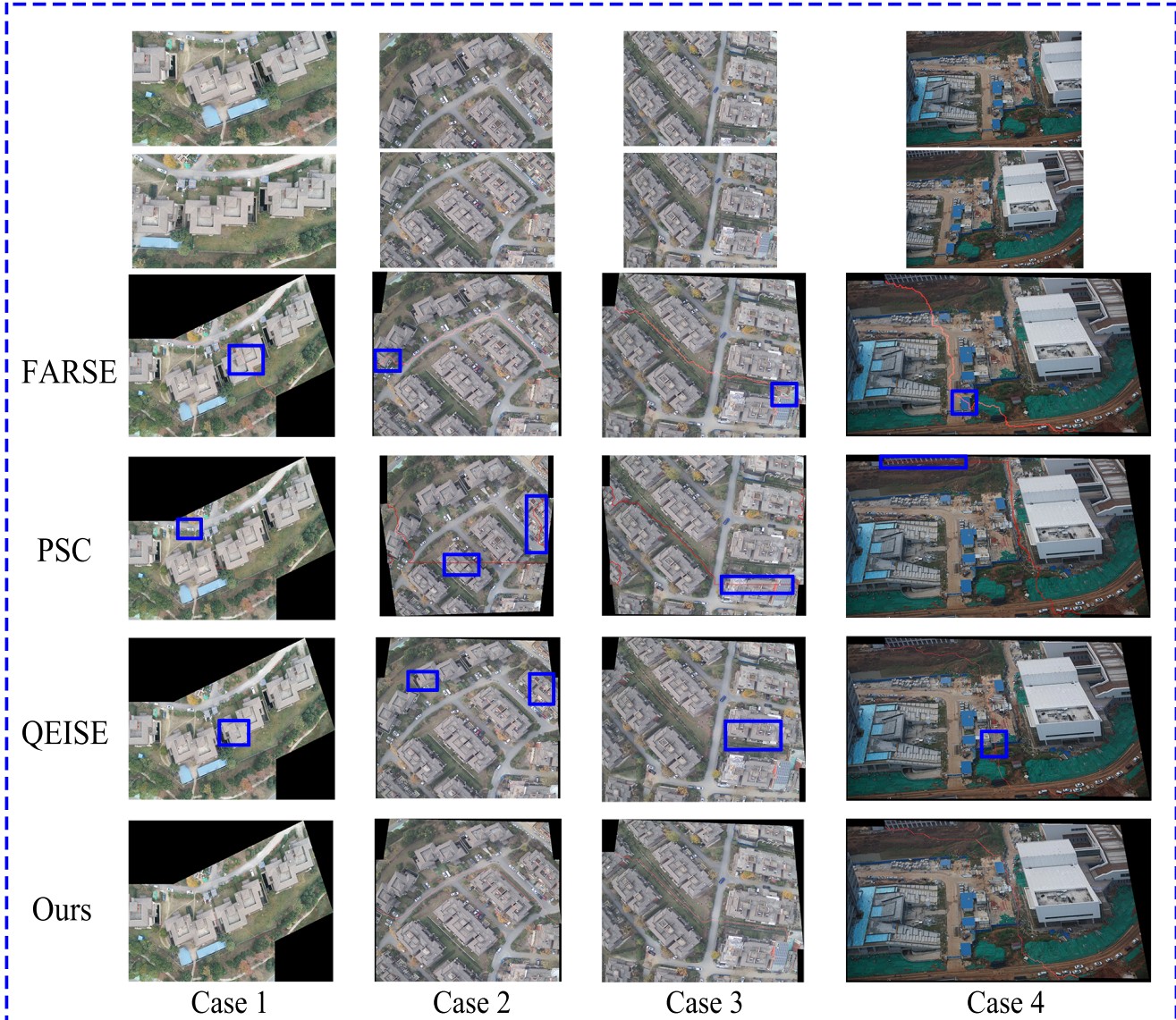

**Figure 7.** Seam location compared to other seam algorithms. The first and second lines are the input images. The third, fourth, fifth and sixth lines are the results of FARSE, PSC, QEISE and our seam algorithm. Seams are indicated in red. Blue boxes highlight some special structural objects. The percentage of overlap between two images in these four cases is 52.74%, 62.98%, 78.08%, 76.07%, respectively.

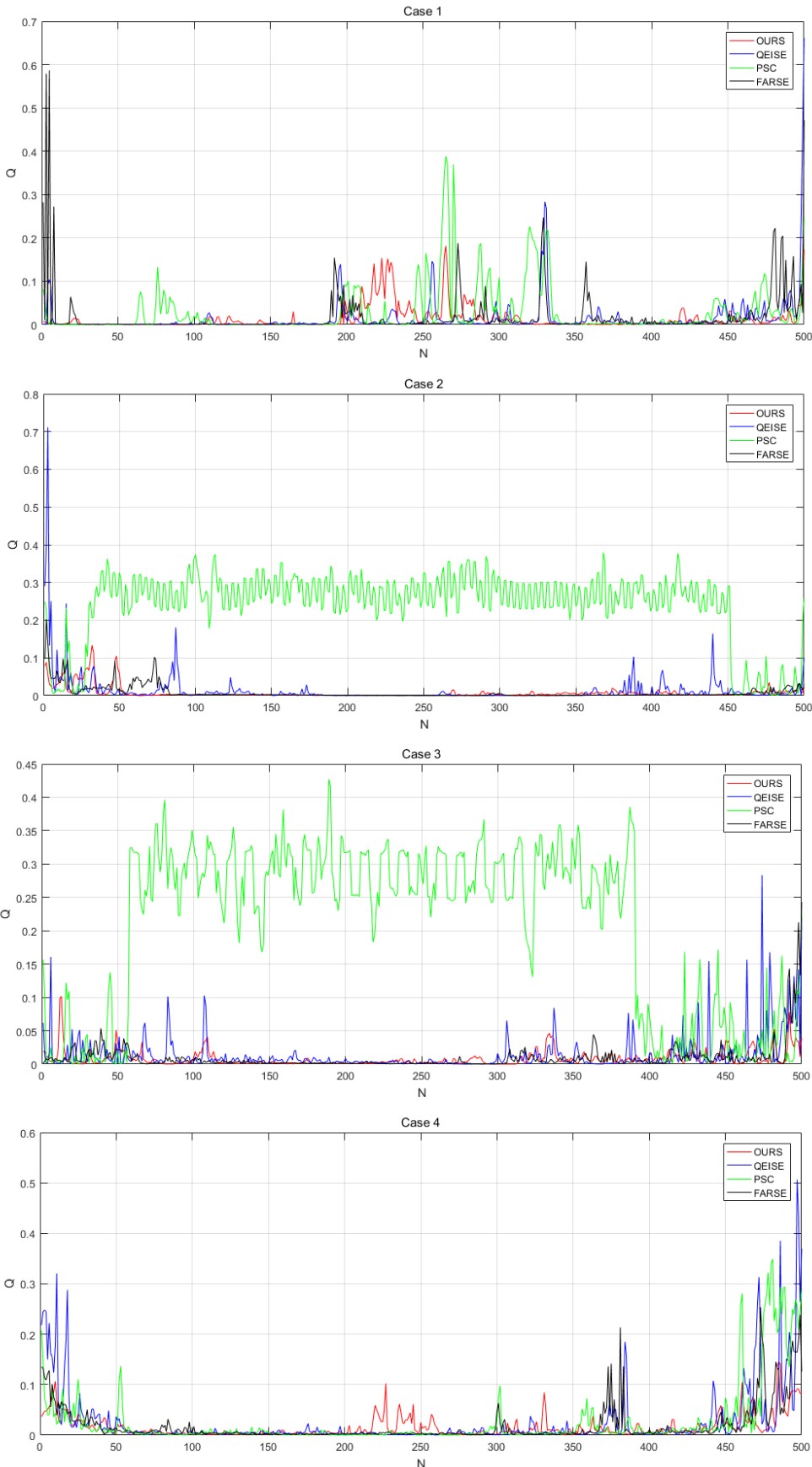

**Figure 8.** Quality curve comparison of seams with different seam algorithms.

**Table 1.** $Q_{PSNR}$ of seams with different algorithms (dB). Bold indicates the best results.

|        | FARSE   | PSC     | QEISE   | Ours        |
| ------ | ------- | ------- | ------- | ----------- |
| Case 1 | 51.9395 | 46.2676 | 52.4685 | **53.1513** |
| Case 2 | 49.0071 | 35.1541 | 49.4543 | **51.8483** |
| Case 3 | 47.0600 | 34.7118 | 48.9986 | **50.3483** |
| Case 4 | 51.1492 | 49.1601 | 50.5044 | **53.3715** |

**Table 2.** $Q_{SSIM}$ of seams with different algorithms. Bold indicates the best results.

|        | FARSE  | PSC    | QEISE  | Ours       |
| ------ | ------ | ------ | ------ | ---------- |
| Case 1 | 0.0176 | 0.0276 | 0.0141 | **0.0133** |
| Case 2 | 0.0079 | 0.2391 | 0.0131 | **0.0073** |
| Case 3 | 0.0088 | 0.2039 | 0.0135 | **0.0080** |
| Case 4 | 0.0186 | 0.0253 | 0.0244 | **0.0154** |

**Table 3.** Time consumed with different algorithms (s). Bold indicates the best results.

|        | FARSE | PSC   | QEISE | Ours     |
| ------ | ----- | ----- | ----- | -------- |
| Case 1 | 3.12  | 12.26 | 20.24 | **7.24** |
| Case 2 | 2.70  | 7.16  | 9.05  | **6.31** |
| Case 3 | 2.59  | 5.24  | 5.33  | **4.14** |
| Case 4 | 2.36  | 7.46  | 6.535 | **4.68** |

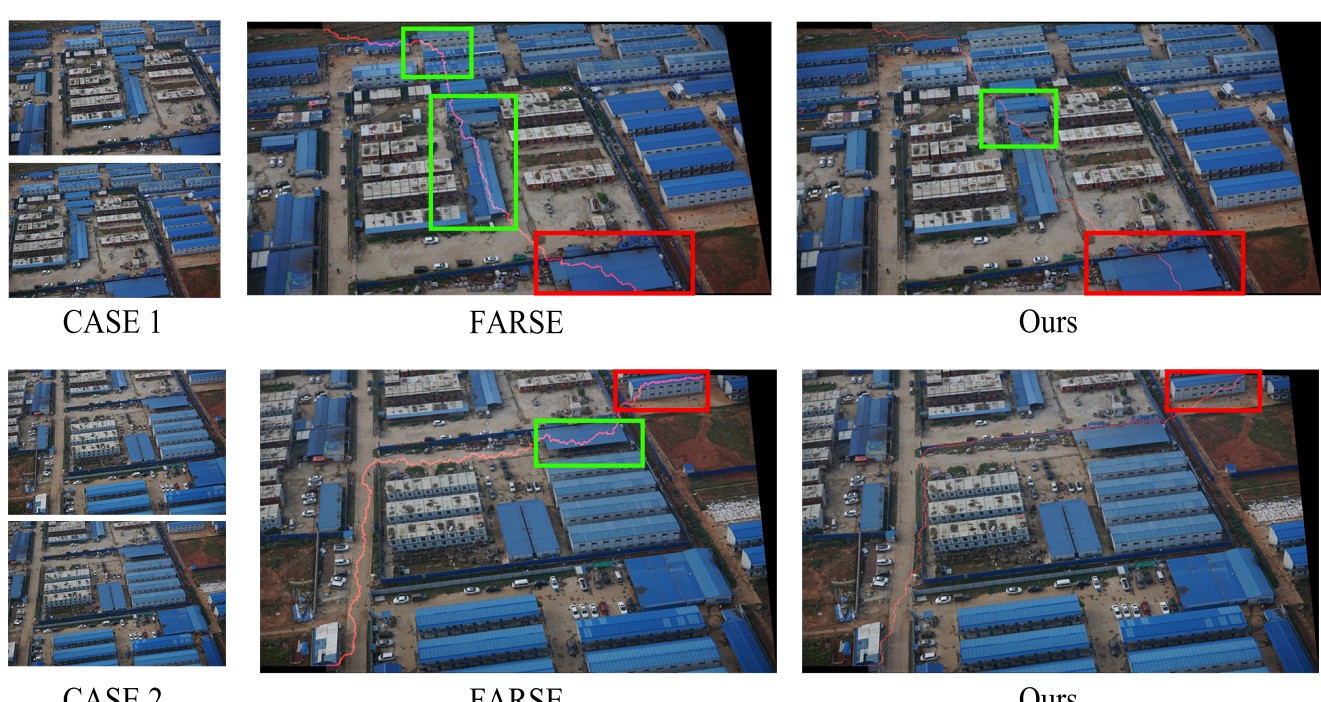

CASE 1 · FARSE · Ours

CASE 2 · FARSE · Ours

**Figure 9.** The comparative experiment between our seam algorithm and FARSE in the case of dense houses. The red line indicates the seam. Red and blue boxes highlight some special structural objects. The percentage of overlap between two images in these two cases is 74.43%, 86.62%, respectively.

### 4.3. Shape Correction

In this paper, the half-projective warp combined with the seam algorithm is proposed to solve the problem of shape distortion retained in stitched images. The half-projective warp proposed by SPHP uses projection transform in the overlapping region and similarity transform in the non-overlapping region. After using the seam algorithm to obtain the cut image, we fix the cut reference image and apply the half-projective warp to the original

target image. This is equivalent to increasing the non-overlapping region and reducing the overlapping region. Then, the parameters $u_1$ and $u_2$ are determined according to the position of the seam to construct the warp. The main distorted regions are concentrated in the overlapping region of the target image. However, we cut and remove the region according to the position of the seam. Finally, we obtain more images from non-overlapping region similarity transformation to form our stitching results.

As mentioned above, UAV images with large parallax are unable to obtain accurate registration, and the objects in the overlapping area will produce shape distortion after transformation. Figure 10 shows several stitching cases. We can see that, on the UAV image stitched according to the seam, some houses will have inclined tensile deformation (indicated by the blue boxes), which is not in line with our human sensory cognition. In our method, the houses originally in the overlapping area are classified into non-overlapping areas, and the original angle and direction are maintained after similar transformation. Finally, our stitching results retain the shape of the original image as much as possible and obtain better visual effects.

In particular, we compare the results with those obtained by combining SPHP and our seam algorithm. We use SPHP instead of global homography for our preliminary registration, and then use our seam algorithm to obtain the stitching image. As shown in Figure 11, since some houses are in the overlapping area at the beginning, SPHP cannot solve the problem of shape distortion of houses in the overlapping area. As indicated by the blue box, the house still tilts and may produce more deformation. Our method can divide the non-overlapping areas through the position of the seam, which can cause more regions to undergo similar transformation and maintain the original shape of the object.

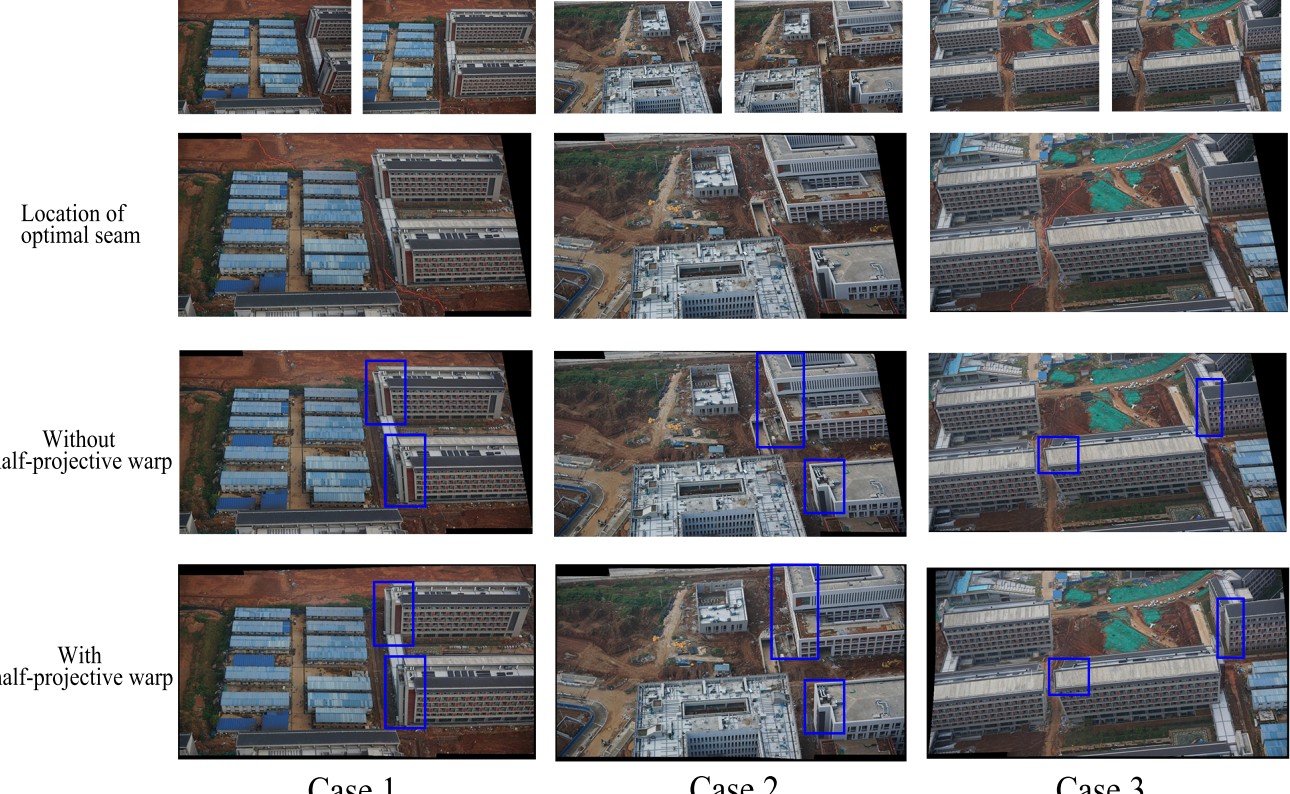

Location of optimal seam

Without half-projective warp

With half-projective warp

Case 1　　　　　　　　Case 2　　　　　　　　Case 3

**Figure 10.** Shape preserving effect of half-projective warp combined with seam position. The first line is the input images. The second line is the location of our seam. The third line is the images without shape preservation. The fourth line is the shape preserved images. The red line indicates the seam. Blue boxes highlight some special structural objects. The percentage of overlap between two images in these three cases is 68.04%, 72.03%, 75.79%, respectively.

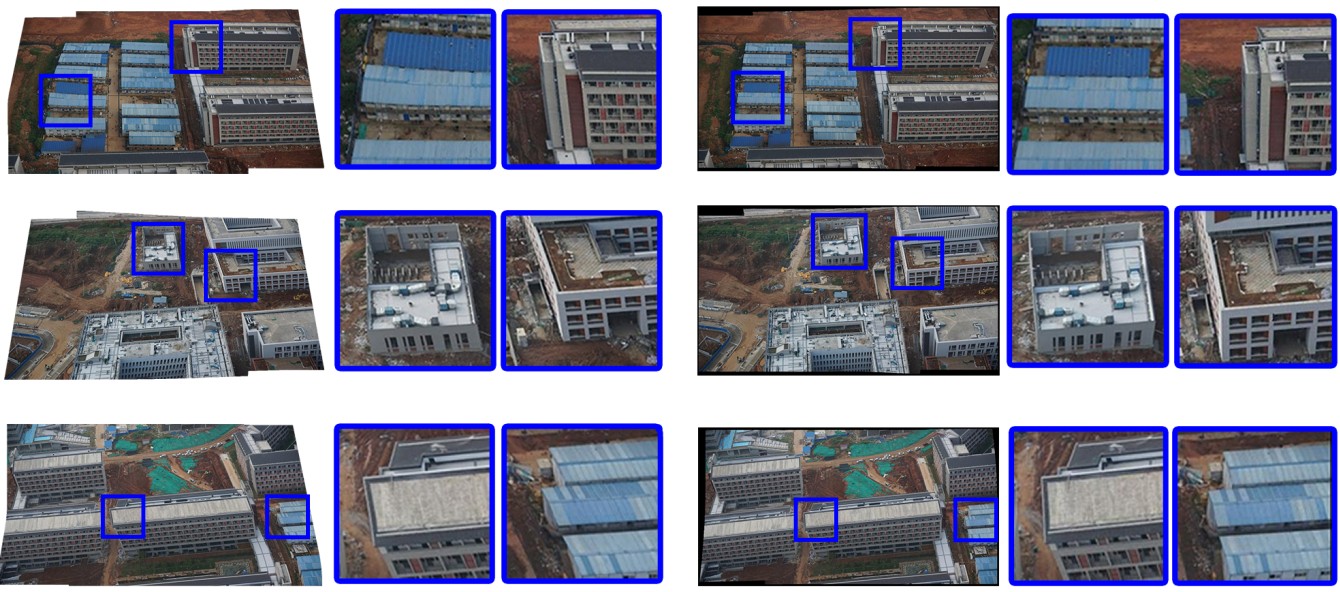

SPHP + our seam algorithm

Ours

**Figure 11.** The comparative experiment between SPHP and our seam algorithm and our method. Blue boxes highlight some special structural objects.

### 4.4. Extended Experiment

We apply the proposed method to hyperspectral images and compare it with automatic stitching for hyperspectral images using robust feature matching and elastic warp (AHREW) [44]. As shown in Figure 12, neither AHREW nor our method has ghosting and blurring. The difference is that AHREW relies on accurate alignment, so it produces angle distortion in non-overlapping areas, as shown by the red box. The seam searched by our method can still avoid structural objects, and can reduce the angle distortion of non-overlapping areas.

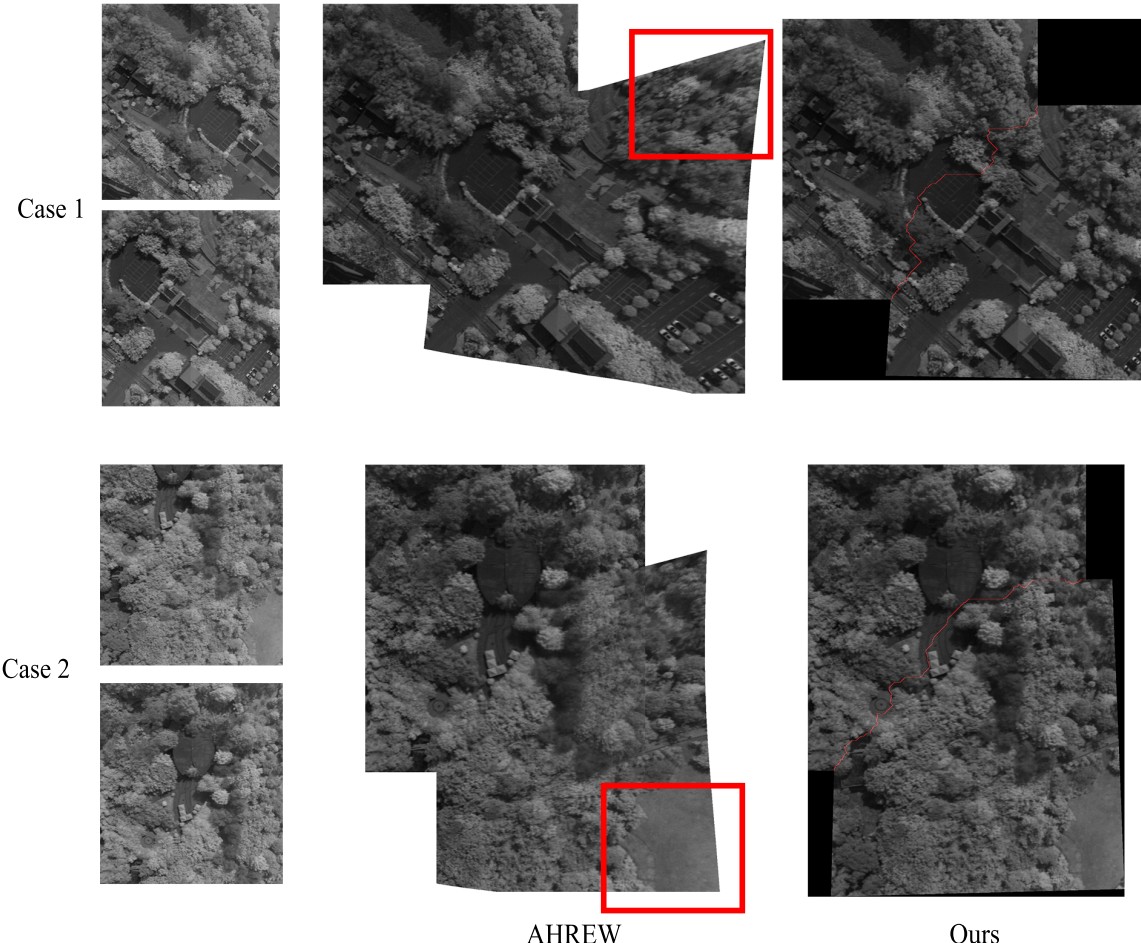

Case 1

Case 2

AHREW                                    Ours

**Figure 12.** The comparative experiment between AHREW and our method on hyperspectral images. The red line indicates the seam. Red boxes highlight angle distortion areas on the results of AHREW.

## 5. Discussion

Due to the limited display range of images taken by a single UAV lens, a method is required in the field of remote sensing to stitch adjacent images. UAV images have the characteristics of large parallax, and it is difficult to achieve accurate registration, which leads to ghosting and blurring of the images. Previous studies have demonstrated that the use of seam stitching algorithms can reliably eliminate ghosting and blurring. Although these studies have revealed some important findings, there are also shortcomings. The seam algorithm usually searches for seams at the cost of the difference reflecting the similarity between the overlapping regions of the two images. Color differences and structural differences are commonly used to describe the cost of differences. Few studies have considered how to use the different characteristics of the rich information contained in UAV images to constrain the path of the seams. In addition, another problem with the seam algorithm is that the distortion or viewing angle distortion caused by the image registration will be preserved.

In order to solve the above shortcomings, we first define a new difference cost that can better constrain the seam search. We use the high-frequency part of the image to construct structural differences to increase the probability that the seam path is in the flat area of the image. The structure object has a large amount of line information, and the line difference is added to the difference cost to reduce the possibility of the seam passing through the structure object. Under the condition that the seam can be searched, we further restrict the search range of the seam. Then, according to the difference cost that we defined, a seam algorithm that can find the smallest cost difference is proposed. Different experimental

results show that our method can solve the problem of ghosting and blurring in stitched images (Figures 4–6). Compared with other advanced seam algorithms, our seams can better avoid structural objects (Figure 7). This leads to a better evaluation than other seam algorithms on the defined metrics (Tables 1–3, Figure 8). The advantage of the seam algorithm is that it can avoid ghosting and blurring of stitched images, but it also has some defects. When one end of the seam appears on an object, the seam will inevitably pass through the object, which can easily cause visual error. When structural objects in UAV images are very dense, it will increase the difficulty of searching for suitable seams. Compared with other algorithms, our proposed seam algorithm can better avoid passing through structural objects and obtain better seams when processing the image stitching of dense structural objects (Figure 9).

In the previously mentioned SPHP method, applying similar transformations in non-overlapping areas can solve the perspective distortion and retain more source image information, but it cannot solve the problem of distortion in the overlapping areas. We found that, after cutting the image according to the seam, using the cut part as the new overlapping area, applying the SPHP method to register the image can successfully solve the distortion problem of the overlapping area. After this, the non-overlapping area is still retained by applying similarity transformation, while the overlapping area is removed according to the seam. This is equivalent to retaining more non-overlapping area information and removing the distorted overlapping area. The experimental results (Figures 10 and 11) show that our method can retain more original image information and improve the perception of stitched images. In addition, our proposed method can be used to stitch hyperspectral remote sensing images (Figure 12), and has good application prospects.

## 6. Conclusions

In this article, we propose a method for UAV image stitching based on the optimal seam algorithm and half-projective warp. The main purpose of this method is to obtain a natural panoramic image with a good visual effect and no ghosting or blurring. In the seam algorithm, we propose a new definition of the difference matrix and restrict the region of seam search. Then, we propose a seam search algorithm based on global energy minimization, which causes the seam to avoid structural objects and move along the flat area. Finally, according to the position of the seam and combined with the half-projective warp, more areas retain the original shape, so as to improve the sensory effect of the stitched image. Experiments show that our method exceeds popular methods, and our method is also suitable for hyperspectral remote sensing image stitching. Our future research will continue to focus on more efficient seam search algorithms.

**Author Contributions:** Conceptualization, J.C. and Z.L.; methodology, J.C. and Z.L.; software, Z.L.; validation, Z.L.; formal analysis, J.C., Z.L., C.P., Y.W. and W.G.; resources, J.C. and C.P.; data curation, J.C.; writing—original draft preparation, Z.L.; writing—review and editing, J.C., C.P., Y.W. and W.G.; visualization, Z.L.; supervision, J.C. and C.P.; project administration, J.C.; funding acquisition, J.C. and C.P. All authors have read and agreed to the published version of the manuscript.

**Funding:** This work was supported in part by the National Natural Science Foundation of China, nos. 62073304, 41977242 and 61973283, and in part by the China Postdoctoral Science Foundation, no. 2021M702533.

**Conflicts of Interest:** The authors declare no conflict of interest.

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
