# Peer review of "UAV Image Stitching Based on Optimal Seam and Half-Projective Warp"

_remotesensing, doi:10.3390/rs14051068_

Round 1

Reviewer 1 Report

Dear Authors,

Thank you for your contribution. The presented topic is interesting, especially digital image processing and photogrammetry. Results of your method can be used both by amateurs and professionals.

However, I have some remarks about the presented research. First, I am missing information about the overlap between used images. Have you verified your method for different overlap percentages? Second, I am missing data about viewing angles for images presented in 4.3. Of course, the difference can be easily observed, but what are the values. What will happen when the angle difference is greater?

Could you define “large parallax”- that term is often used in your research.

The article is very well structured. The method descriptions are concise and sufficient, similarly to the methodology of the original seam algorithm. I have some remarks of text and figures editing. First, rewrite the first paragraph in the Introduction- in my opinion, it is tough to read.

Next, in Figure 1, in my opinion, it would be more appropriate to make a workflow instead of putting separate images with letters and descriptions. The same with Figure 2. Figure 4 and Figure 5- please add names of methods to images. It is not clear enough which results represent a specific method.

Reviewer 2 Report

The paper proposes a stitching strategy based on optimal seam and half-projective warp for UAV images. Looking only at the paper shows that it is well-written in both technical, structure and presentation and also based on the English language, however, based on the name of the special issue, Advances in Hyperspectral Remote Sensing: Methods and Applications, I assumed that the stitching process could have been done on a HSI dataset domain, like the one in 

Y. Zhang, Z. Wan, X. Jiang, and X. Mei, “Automatic stitching for hyperspectral images using robust feature matching and elastic warp,” IEEE J.
Sel. Topics Appl. Earth Observ. Remote Sens., vol. 13, pp. 3145–3154,
2020

Peng, Zongyi, et al. "Hyperspectral Image Stitching via Optimal Seamline Detection." IEEE Geoscience and Remote Sensing Letters, vol. 19, pp. 1-5,2021

etc. and compare their results with at least one of them. The authors are recommended to mention the word "Hyperspectral" in the manuscript.

The paper seems to be an extended version of the older "Half-Projective Warps" [26] so that it should describe it more and explain how it goes from Computer Vision and Pattern Recognition, which [26] has been published in, to hyperspectral remote sensing.

Anyway, the author used PSNR and SSIM as two quantitative measures for the comparison of their method with other state-of-the-art approaches. How can you add about the complexity of the methods? And the time consumed on the process. 

Throughout the paper, the authors have mentioned "optimal seam algorithm" I recommend to authors to describe it earlier in the beginning of the paper that what is meant by the term "optimal". In fact,  based on what criteria (or in other words, loss function) this algorithm is optimal compared to the previous sub-optimal approaches.

Reviewer 3 Report

Summary

This paper proposes a UAV image stitching method based on optimal seam algorithm and half-projective warp. Its main contribution consists in effectively retain the original information of the image using optimal seam and half-projective warp.

Broad comments

The document is relatively easy to read and follow.

The English needs some review.

The document is well supported with references although old.

The subject of the paper is interesting and with a great potential of application.

One of the weaknesses of this study is that the proposed algorithm requires that the images to be stitched contain flat areas since it is based on a seam search algorithm with global energy minimization which makes the seam avoid structural objects and move along the flat area. Authors should have presented also images where flat areas are unavailable in order to gauge the performance of the proposed algorithm in those situations when compared with the presented state-of-the-art algorithms.

Specific comments

At the beginning of the Introduction section please rephrase “With the development of economy, various advanced UAVs equipped with complete photography equipment.”

In the Introduction section please rephrase “the current UAV is difficult to obtain large-area and high-resolution observation images”

In the Introduction section please rephrase “and so on [8].”

In the Introduction section please rephrase “have been applied to image stitching and achieved great effect”

In Related Work section please rephrase “improve alignment ability can solve some small parallax problems”

In Related Work section please rephrase “proposed a feature refinement model based on Bayesian theory is proposed”

Caption of Figure 3 should start with capital letter. Please correct.

In subsection 4.2 please correct “seam algorithms to prove the our effectiveness”

Since Figure 10 is referred in the text before Discussion section it should be moved to that location.

In the Conclusions section please correct “This paper presents a UAV image stithcing method based”

Reviewer 4 Report

This paper presents an image stitching method for remotely sensed images. The novelties lie in the optimal seam algorithm and the half-projective warp. After carefully examining the manuscript, the reviewer has some assessments, as follows:

  1. The authors used many abbreviations without providing their full names, such as UAV, SIFT, RGB, and LSD, to name but a few. It is a common practice to write the full name in the first instance before using the corresponding abbreviation, even for the well-known RGB. Thus, please proofread the entire paper and provide the full name where it was absent.
  2. It is highly advisable to use thicker color lines and boxes for highlighting in Figures 1, 2, 4, 5, 6, and 7. The current thickness is too small to be noticeable.
  3. Regarding the optimal seam algorithm, the authors should provide details of what the term “optimal” refers to. For example, it should be clear whether the optimal seam algorithm results in the smallest runtime or yields the optimum seam (the one close to the global minimum in the search space).
  4. The description of the optimal seam algorithm in Section 3.1 is somewhat vague because it is not easy to figure out how the algorithm works from the textual description. For example, it is suggested to go into detail about the process of finding the threshold e and then deriving the search region R. In the reviewer’s humble opinion, it would be better to tabulate the pseudocode and provide a flowchart.
  5. As the optimal seam algorithm is one of the novelties, it is essential to
    • discuss its pros and cons, and
    • its difference from existing seam algorithms.
  6. Please take care with the consistency of symbols in equations and the main text. For example, the homography matrix H mentioned in Section 3.2 should be boldfaced to be consistent with its representation in Section 3.1. Also, the cost function in Equation (19) must be denoted by another letter, because E is used for the difference matrix in Equation (11).
  7. The authors should discuss how to find f1, f2, f3, f4, s1, s2, s3, and s4 in detail. This piece of information is essential for reproducibility.
  8. In Section 4.1, the latest among four benchmark algorithms is dated back to 2017. Thus, it is highly advisable to change the phrase “most advanced alignment stitching algorithms” to “popular alignment stitching algorithms” for a more accurate description.
  9. The paper is not well-formatted according to the MDPI’s guidelines. Therefore, please review the guidelines and re-format the paper.

Round 2

Reviewer 1 Report

Thank you for applying my all suggestions. 

Author Response

Thanks for the reviewer's positive feedback.

Reviewer 2 Report

The concerns in the previous round of peer review have been addressed so I suggest accepting the paper in the current form.

Author Response

(The authors gave the same response as above.)

Reviewer 4 Report

The reviewer highly appreciates the authors’ effort in revising the manuscript as most of comments from the first round have been raised sufficiently. Therefore, this time, the reviewer only has some minor remarks as follows:

  1. The pseudocode in Algorithm 1 should be amended for clarity and readability. From the reviewer’s point of view, lengthy descriptions in the pseudocode are confusing, posing significant difficulties for interested readers to reproduce the results. Hence, it is advisable to use notations closely related to the authors’ favorite programming language (e.g., C/C++ or Python). To be more specific, the authors can consider the following:
    • using set notations to define the input and output (this type of definition provides information about the data type and the number of elements), and
    • defining some auxiliary functions (e.g., sorting) to replace the lengthy description in steps 1, 5, 6, and 12.
  2. The manuscript should be proofread carefully to avoid incorrect references within the paper. For example, it is deemed that Equation (15) is derived by substituting Equations (13) and (14), not (9) and (10), into Equations (2) and (3).
  3. If the authors prepare the manuscript using LaTeX, it is advisable to use “soft” cross references to avoid the above problem. Also, please use \eqref{} instead of \ref{} when referring to equations.

Author Response

Q1: The pseudocode in Algorithm 1 should be amended for clarity and readability. From the reviewer’s point of view, lengthy descriptions in the pseudocode are confusing, posing significant difficulties for interested readers to reproduce the results. Hence, it is advisable to use notations closely related to the authors’ favorite programming language (e.g., C/C++ or Python). To be more specific, the authors can consider the following:

  • using set notations to define the input and output (this type of definition provides information about the data type and the number of elements), and
  • defining some auxiliary functions (e.g., sorting) to replace the lengthy description in steps 1, 5, 6, and 12.

R: Thanks for this suggestion. We have modified Algorithm 1 and simplified the lengthy description as much as possible.

Q2: The manuscript should be proofread carefully to avoid incorrect references within the paper. For example, it is deemed that Equation (15) is derived by substituting Equations (13) and (14), not (9) and (10), into Equations (2) and (3).

R: Thanks for pointing out this problem, and we have corrected the error. 

Q3: If the authors prepare the manuscript using LaTeX, it is advisable to use “soft” cross references to avoid the above problem. Also, please use \eqref{} instead of \ref{} when referring to equations.

R: Thanks a lot for the detailed suggestions. We have used “soft” cross references to modify our paper.